# An insight into the gut microbiota of healthy and allergic West Highland Whiter Terrier dogs

**Vera Felten**[1], **Jonathan Louis Turck**[2], **Stefan Unterer**[1], **Claude Favrot**[1], **Jan Suchodolski**[2], **Nina Maria Fischer**[1☯], **Ana Rostaher**[1,3☯*]

**1** Clinic for Small Animal Internal Medicine, Vetsuisse Faculty, University of Zurich, Zurich, Switzerland, **2** Department of Small Animal Clinical Sciences, College of Veterinary Medicine and Biomedical Sciences, Texas A&M University, College Station, Texas, United States of America, **3** Small Animal Hospital, School of Veterinary Medicine, University of Glasgow, Glasgow, United Kingdom

☯ These authors contributed equally to this work.
* ana.rostaher@glasgow.ac.uk

## Abstract

Canine atopic dermatitis (cAD) is a multifactorial, genetically predisposed inflammatory skin disease, with a high prevalence in many different dog breeds, with certain breeds being particularly affected, like the West Highland White Terrier (WHWT). While gut microbiome alterations are linked to allergic diseases in humans and some dog breeds, this relationship has never been investigated in WHWT dogs. This study aimed to compare the gut microbiome of allergic and healthy WHWT dogs to explore its role in cAD. Fecal samples from 21 WHWT dogs (12 allergic, 9 healthy) were analyzed using DNA shotgun sequencing and qPCR assays. No significant differences were observed in alpha or beta diversity, and no significant abundance of bacterial taxa was identified between allergic and healthy dogs. The Dysbiosis Index (DI) did not differ between the allergic (median −4.2, range −6.6 to 1.3) and healthy group. However, a moderate negative correlation was found between the relative abundance of *E. coli* and pruritus severity. These findings indicate that while the gut microbiome overall may not significantly contribute to cAD pathogenesis in WHWT dogs, some species like *E. coli* may be associated with clinical symptoms. Further studies are needed to investigate this relationship and to explore the role of genetics and the gut microbiota across different breeds with a larger number of dogs and with multi-omics approaches.

## Introduction

Canine atopic dermatitis (cAD) is a clinical syndrome characterized as inflammatory and pruritic skin disease, resulting from a combination of genetic predispositions and environmental factors [1,2]. Humans also suffer from atopic dermatitis (AD), and there are many similarities between canine and human atopic dermatitis in terms of pathogenesis, clinical signs and treatment strategies [3,4]. Epidermal barrier

---

**Data availability statement:** All relevant data are publicly available from the Zenodo repository (https://doi.org/10.5281/zenodo.15849870).

**Funding:** This study was financially supported by the Swiss National Science Foundation (https://www.snf.ch) in the form of a grant (215128) received by AR and NMF. This study was also financially supported by the American Kennel Club Canine Health Foundation (https://www.akcchf.org) in the form of a grant (03162) received by AR and NMF. This study was also financially supported by the Westie Foundation of America (https://westiefoundation.org) in the form of an award received by AR and NMF. This study was also financially supported by the Purina PetCare Research Excellence Fund (https://www.purina.com) in the form of an award received by JS. The funders had no role in study design, data collection and analysis, decision to publish, or preparation of the manuscript.

**Competing interests:** The authors have declared that no competing interests exist.

impairment, dysregulated cell-mediated immune response, and high serum IgE levels against environmental allergens are observed in affected individuals of both species [3,4].

Canine AD closely mirrors its human AD counterpart in terms of its spontaneous onset, the nature and distribution of cutaneous lesions, the occurrence of immune dysregulation, impairments in the skin barrier, and the common occurrence of secondary infections that complicate the condition [5]. Furthermore, the long-standing cohabitation of dogs with humans has resulted in shared environmental and lifestyle factors, which in turn contribute to a shared exposome that influences the risk of developing atopic dermatitis [6]. Therefore, both species offer valuable reciprocal AD models for the study and development of management and preventive strategies for this disease.

In humans the prevalence of atopic dermatitis has increased in the last years and is estimated to be between 10-20% worldwide [7]. The exact prevalence in dogs remains uncertain, however, it is assumed that about 10% of the global dog population is affected [8]. Certain dog breeds are notably predisposed to a higher prevalence of canine atopic dermatitis, with the West Highland White Terrier (WHWT) being among the most affected [8,9]. Favrot et al could show in their study that in more than half of the included WHWT dogs, living in Switzerland, the first symptoms of cAD first appeared before they reached the first six months of age [9]. A genetic and hereditary component in WHWT dogs has been discussed in several studies, with evidence pointing to specific genes that may contribute to the increased predisposition to cAD in this breed [10–14]. While genetic predisposition plays a crucial role in the development of canine atopic dermatitis in breeds such as the WHWT, genetics alone cannot fully account for the pathogenesis of the disease. Therefore, it is essential to consider additional factors playing a role in the pathogenesis in this disease to achieve a more comprehensive understanding and to identify possible new approaches for prevention and treatment.

As a key component of the cutaneous barrier, the skin microbiome and its disruptions, known as cutaneous dysbiosis—characterized by an imbalance in the microbial population, a decrease in microbial diversity, and a reduction in beneficial bacteria—have garnered significant attention over the past decades. Numerous studies have shown that both dogs and humans affected by AD experience cutaneous dysbiosis [15–18]. However, it is still uncertain whether this dysbiosis is a cause or an effect of the atopic condition. Moreover, there is evidence from a previous study involving WHWT dogs indicating that the early-life skin microbiota of puppies involved in this cohort study was not associated with development of AD [15]. Hence, there is a growing emphasis on the gut microbiome, which could play an additional or even a more critical role in the development of allergies in dogs.

The "gut-skin axis" has been supported by several studies, which demonstrate that alterations in gut microbiota can significantly impact skin health and contribute to the development of skin conditions such as atopic dermatitis [19,20]. Previous studies showed that the compositional differences in the gut microbiome are associated with the development and severity of atopic dermatitis through immunologic, metabolic, and neuroendocrine pathways [20].

The gut microbiota directly influences T cells, and it is known that short-chain fatty acids (SCFAs), produced by certain bacteria, can have an anti-inflammatory effect and influence the immune system [20,21]. The SCFAs generated by gut microbiota can trigger SCFA-sensing G-protein coupled receptors (GPCRs) and/or hinder histone deacetylases (HDACs). This initiates a series of signals that help to reduce inflammation and re-establish the balance between TH1 and TH2 responses. Additionally, the microbial metabolite D-tryptophan has the potential to restore this balance as well [22]. Several studies in human research focus on identifying the bacteria responsible for the development and maintenance of atopic dermatitis, as well as categorizing these bacteria into protective and triggering factors related to the disease [23,24].

In veterinary medicine, the importance of a healthy gut microbiome for overall - health is well established. There is clear evidence that dysbiosis plays a major role in several intestinal diseases in animals but data on the role of gut microbiota in cAD is still scarce [25–27]. However, in the few studies performed, there is evidence that the gut microbiome differs between allergic and healthy dogs [26,27]. Based on this preliminary data and the large knowledge gap, there is a clear need for further research to better understand how gut microbiome may influence the development and progression of allergic skin conditions in dogs. In this study, we aimed to compare the gut microbiome of allergic and healthy dogs from a high-risk AD breed, specifically the WHWT breed, to gain insights into the disease's pathogenesis and the gut microbiome's role therein.

## Materials and methods

### Study population

A total of 21 WHWT dogs were prospectively recruited during October 2023 and June 2024 for this study, comprising 12 dogs with AD and 9 healthy controls. Nineteen of these 21 dogs were prospectively selected from a larger group enrolled in a prior observational birth cohort study [9]. In this earlier study, 107 WHWT puppies from 29 litters from 17 breeders in Switzerland were recruited. All puppies were clinically examined at 6–8 weeks of age and followed minimally three years of age to assess the development of AD. The two-remaining client-owned dogs (one healthy and one allergic) were presented to the Department of the Clinic for Small Animal Internal Medicine at the Vetsuisse Faculty, University of Zurich during routine consultations. All new owners signed a written informed consent form before participating the study, which was approved by the Swiss Cantonal Veterinary Office and conducted in accordance with guidelines established by the Animal Welfare Act of Switzerland (National No. 35805, Cantonal No. ZH069/2023).

### Allergy status assessment

The allergy status of each dog was determined using a structured questionnaire completed by the owners and confirmed through clinical examination by a veterinarian. This comprehensive approach ensured accurate classification of allergic and healthy subjects. The questionnaires to identify allergic and healthy dogs used for this study were adapted from previous studies identifying risk factors for the development of cAD [9,28]. The inclusion of AD cases was based on the following: veterinarian-based AD diagnosis, owner-reported presence of typical AD symptoms (e.g., erythema, alopecia and excoriations) on typical predilections sites (e.g., ears, feet, axillae) associated with pruritus and unresponsiveness to adequate ectoparasite and antimicrobial treatments. The inclusion criteria for the healthy control dogs were absence of veterinarian-based AD diagnosis and other immunological disorders and the age above 5 years. The exclusion criteria for healthy dogs were food allergy improving on exclusion diet, having gastrointestinal problems, moist eczema skin lesions (hot spots), ear infections and inflammation, frequent vomiting, pruritus or were allergy tested. The intensity of itching was evaluated using the Pruritic Visual Analog Scale (PVAS) [29] allowing owners to quantify the severity of their dog's itching on a standardized scale. All data were anonymized to ensure confidentiality and ethical compliance.

## Sample collection

Stool samples from 12 dogs with spontaneous AD and a control group consisting of 9 healthy dogs were collected by the dog owners directly into PERFORMAbiome-GUT tubes (DNA Genotek, Ottawa, ON, Canada). Upon collection, the samples were promptly shipped to the laboratory, where they were stored at −80°C until further analysis.

## Sample analysis

The fecal samples were collected from both groups and were stored at -80 °C before being shipped on dry ice to the Gastrointestinal Laboratory of Texas A&M University, College Station, Texas. The microbial DNA from fecal samples was extracted and quantified using the DNeasy PowerSoil Pro Kit (QIAGEN Inc, Germantown, Maryland) and the Thermo Scientific NanoDrop Microvolume Spectrophotometers respectively. The Nextera XT DNA Library Preparation Kit (Illumina Inc., San Diego, CA, USA) was used for sequencing library preparation, followed by pooling of the libraries. Afterward, SPRI bead purification and concentration were performed with SpeedBeads Magnetic Carboxylate Modified Particles (Cytiva Life Sciences, Marlborough, MA, USA). The pooled libraries were denatured using NaOH, diluted, and spiked with 2% PhiX. Sequencing was carried out on an Illumina NovaSeq6000 with a 2x150 flow cell to achieve a median of 2 million read pairs at the Diversigen laboratory (New Brighton, MN). The data were converted to FASTQ files and filtered for low-quality (Q-score <30) and short lengths (<50) sequences. Adapter sequences were removed, and all remaining sequences were trimmed to a maximum length of 100 base pairs before proceeding with the alignment process. Raw sequence data were uploaded to the NCBI Sequence Read Archive and analyzed using established pipelines. For taxonomic classification, FASTA sequences were aligned to a curated database containing all representative prokaryotic genomes from the NCBI RefSeq collection, as well as additional manually curated bacterial strains. Alignments were performed at 97% identity and compared with reference genomes. Taxonomy assignment was based on the lowest common ancestor approach, with compatibility to at least 80% of the reference sequences. Operational Taxonomic Units (OTUs) with fewer than one million species-level markers, or with less than 0.01% unique genome region matching and less than 0.1% of the entire genome, were discarded. The normalized and filtered data tables were then imported into QIIME2 for subsequent downstream analysis. Prior to conducting diversity analyses, samples were rarefied to depth of 642000, ensuring uniform depth for analysis.

The qPCR assays quantified total bacteria, Blautia, *Clostridium* (Peptacetobacter) *hiranonis*, *Escherichia coli*, Faecalibacterium, Fusobacterium, Streptococcus, and Turicibacter. The qPCR assays have been described previously as previously described [30]. Briefly, the qPCR assays were performed in the following order: at 95 °C maintained for 2 min, 40 cycles at 95 °C for 5 s, and then annealing at the optimized temperature for 10 s, using 10 μL of SYBR-based reaction mixtures (5 μL of SsoFast™ EvaGreen® supermix [Bio-Rad Laboratories GmbH, Düsseldorf, Germany]), 1.6 μL of high-quality PCR water, 0.4 μL of each primer (final concentration: 400 nM), and 2 μL of DNA. Both positive and negative controls were included for all qPCR assays to ensure the accuracy and reliability of the results. The DI was calculated based on the results of the qPCR assays using a previously described algorithm [30].

## Statistical analysis

Statistical analysis was conducted to test the hypothesis that there is a difference in the fecal microbiome between the two groups. Data were tested against the hypothesis of normal distribution by conducting the Shapiro-Wilk's test using GraphPad Prism v10.0.2 (GraphPad Software, Inc., La Jolla, CA, USA). The data did mainly follow a normal distribution. However, due to the zero inflated and compositional nature of microbiome data, non-parametric methods were applied. Statistics on the Shannon, Chao1, and Observed Features indices were performed using a one-way ANOVA. All alpha diversity statistical analyses were performed in R statistical computing software (R v 4.2.3). To quantify the dissimilarity in microbial structure between groups, a non-parametric ANOSIM (analysis of similarities) was performed on the Bray-Curtis distance matrix in Quantitative

Insights Into Microbial Ecology 2 (QIIME2 2024.2). Groups were defined based on the individual's respective treatment and the time point at collection. To compute p-values, 999 permutations were calculated, and pairwise comparisons were performed against baseline. The significance level was set at a p-value of <.05. Differentially abundant taxa between the groups at phylum, family, genus, species, and strain levels were identified using Analysis of Composition of Microbiomes with Bias Correction (ANCOM-BC) from the R package ANCOMBC (v 2.2.0) using the default parameters. Subsequent analysis was performed with MaAsLin2 (v 1.16.0). Features were normalized with total sum scaling and log transformed. No differentially abundant features were found using ANCOMBC or MaAsLin2. Features were then filtered to exclude any filters with less than 0.001% overall relative abundance. A Wilcoxon signed-rank test was then performed within R. No features were found to be significant following p-value correction. Some selected features are plotted below. A stacked relative abundance table on a phylum level was created utilizing R statistical software v4.2.3 (R Core Team, 2020) with the phyloseq v3.20 package. The Spearman test was used to evaluate the correlations between the abundance of taxa obtained by qPCR and the severity of pruritus.

## Results

### Signalement and clinical data

A total of 21 WHWT were included in the study, of which 12 were in the allergic group and 9 were in the healthy group. The median age of the allergic and healthy dogs in this study was 6.75 (1.5-7.5) and 7 (5-7.8) years, respectively. The male to female sex distribution was 1:2 and was the same for both groups. The median pruritus score in the allergic group was 3 (range: 0–6), measured with the Pruritus visual analogue scale (PVAS) from 0 (low) to 10 (high). In the healthy group, the pruritus score for all dogs was 0. Symptomatic allergy treatment was administered to three dogs in the allergic group, such as oclacitinib (n=1), ear drops or ear cleaner (n=2), allergen-specific immunotherapy (n=1).

### Comparison of bacterial diversity parameters between allergic and healthy dogs

**Sequencing summary.** A total of 21 stool samples from 12 allergic and 9 healthy dogs were analyzed. Following the evaluation of the sequencing reads from the stool microbiota, 2,634 numbers of observations (ASVs) were detected, and 63,292,459 high-quality sequences were obtained. Species richness reached a plateau at approximately 1,100–1,300 observed features with a sequencing depth of 660,000 reads per sample, indicating sufficient capture of microbial diversity.

**Alpha diversity.** Although the median values for Chao1, observed features and Shannon index were lower in allergic dogs as compared to healthy dogs (Fig 1), these values were not statistically significant (p = 0.7). Furthermore, the interquartile range in the allergic group was much broader, indicating a greater variability in microbial diversity within this group.

**Beta diversity.** Beta diversity analyses also revealed no significant differences between allergic and healthy dogs (p = 0.98), as determined by Analysis of similarities (ANOSIM). Principal coordinate analysis by Bray-Curtis did not show clustering of the bacterial gut microbiota between the two groups, supporting the results that there were no significant differences in microbial composition between allergic and healthy dogs (Fig 2).

**Microbial differential abundance.** The composition of the gut microbiota in both groups was divided into the following phyla: Firmicutes, Bacteroidota, Actinobacteriota, Fusobacteriota, Proteobacteria, Campylobacterota and other low abundant phyla. Bacteroidota (median allergic 38.4; Median healthy 34.6) and Firmicutes (median allergic 37.7; median healthy 30.6) were the most abundant Phyla in both groups (Fig 3).

However, no significant differences were observed at the phylum level between healthy and allergic dogs. Similarly, the analysis of the bacterial composition revealed no significant differences between the two groups at the class, order, or family level (Supporting information S1 Fig).

**Dysbiosis index and correlation with severity of pruritus.** The Dysbiosis Index (DI) did not differ between the allergic (median -4.2, range -6.6–1.3) and healthy group (median -4.1, range -6.7–1.0). Only one dog in the allergic group

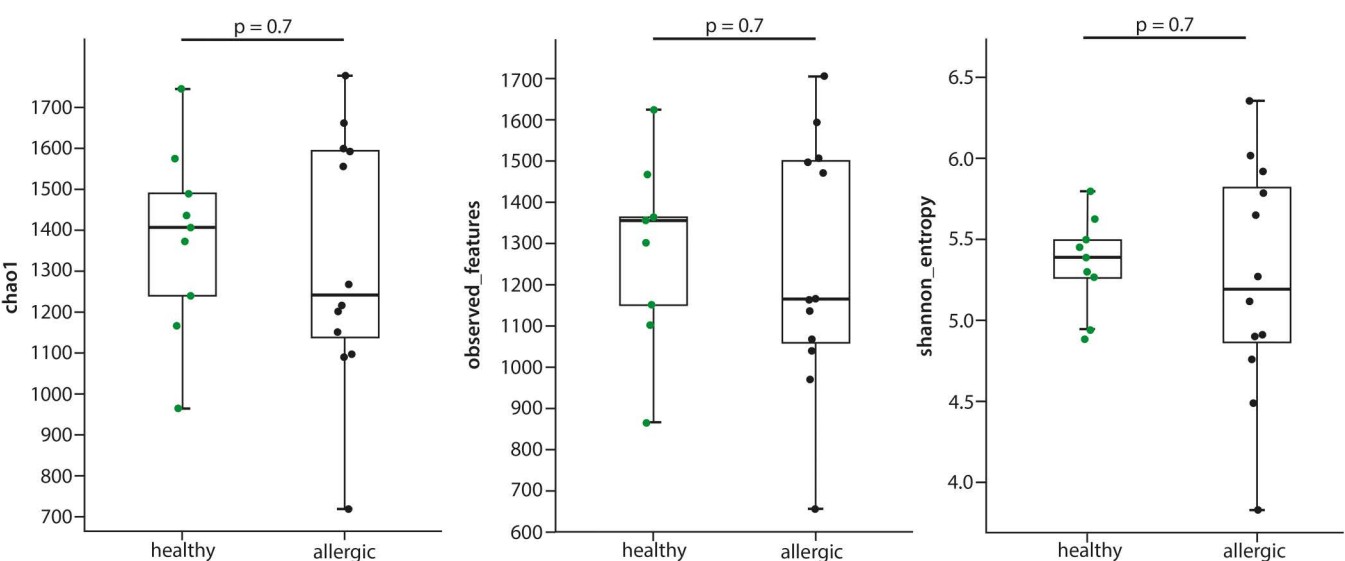

**Fig 1. a-c. Comparison of alpha diversity parameters between atopic and healthy dogs, showing following features.** Chao 1, Observed features, Shannon index was not statistically different between the allergic and healthy dogs.

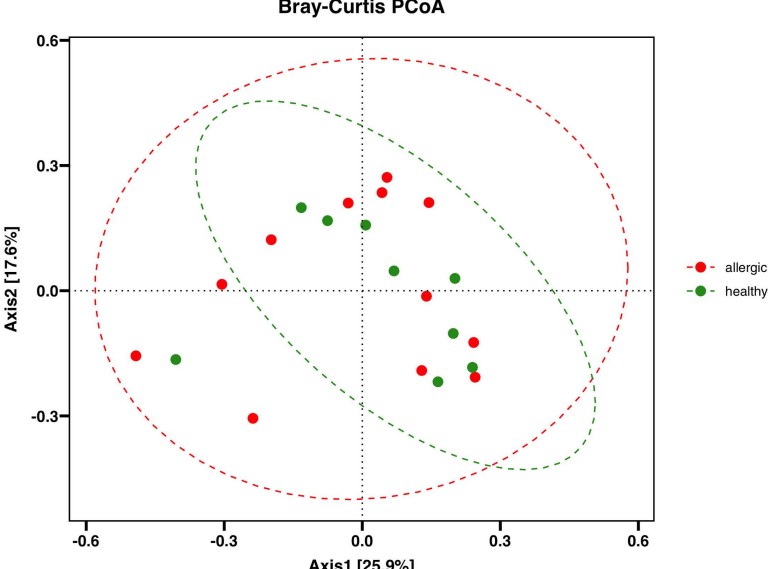

**Fig 2. Principal coordinates analysis (PCoA) for the beta-diversity metrics.** No significant pattern was observed in the beta-diversity metrics between the groups, suggesting similar overall compositions between the allergic and healthy samples.

(n = 12) and one dog in the healthy group (n = 9) showed a mild dysbiosis (Table 1). While no other dog in both groups had a bacterial shift that would indicate a dysbiosis in the gut. We also evaluated the correlation between individual bacterial species represented within DI. *Escherichia coli* showed a moderate negative correlation (R = -0.58, p = 0.049) with pruritus severity (Fig 4).

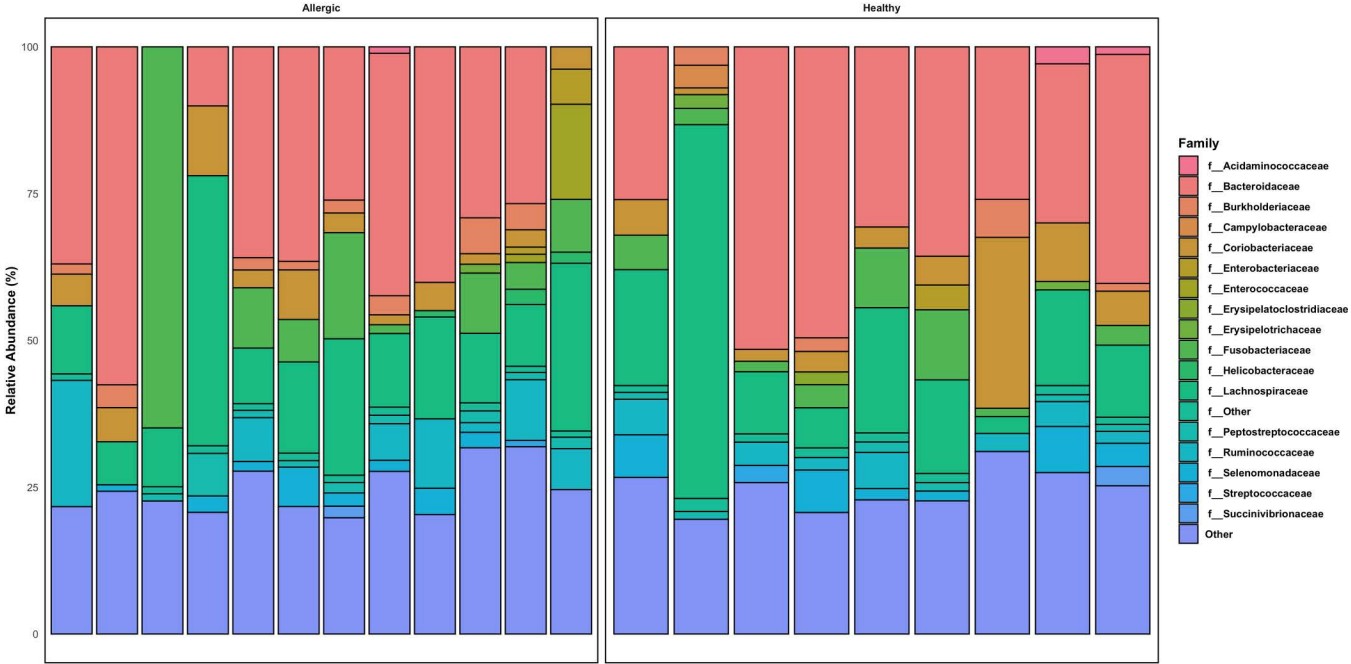

**Fig 3. Relative abundance taxonomy bar plot at the phylum level.** The plot depicts overall phylum abundance between allergic and healthy dogs.

## Discussion

To the best of our knowledge this is the first comparative study to explore the gut microbiome of healthy and allergic WHWT dogs, a well-known high-risk breed for atopic dermatitis. The primary finding of the present study is that there were no significant differences in the fecal microbial composition between healthy and allergic WHWT dogs. Both alpha and beta diversity measures were comparable between the two groups, and no differences were observed at the taxonomic level.

Previous studies have shown significant differences in alpha and beta diversity between healthy and allergic dogs of various breeds [31] and in Beagle dogs [27], with allergic dogs typically exhibiting lower alpha diversity. Conversely, studies on Shiba Inu dogs, another breed at high risk for AD, found no significant differences in diversity between healthy and allergic dogs [26,32]. A previous study has shown that gut microbiome composition can be influenced by the dog breed [33,34]. These inconsistencies highlight that breed-specific factors might not only influence the gut microbiome composition in health but also in disease, such as in canine AD. We hypothesize that the breed effect in WHWT dogs may account for the lack of significant differences observed in our study and that possible changes in the gut microbiota between allergic and healthy dogs may not be strong enough to show significant differences, similar to the findings in the studies with the Shiba Inu dogs.

The composition of the gut microbiota in healthy dogs generally consists of phyla such as Firmicutes, Fusobacteriota, Bacteroidetes, Proteobacteria, and Actinobacteriota [25]. In our study using DNA shotgun sequencing, a method which avoids primer bias, the most abundant phyla detected in the gastrointestinal tract were Firmicutes, Bacteroidota, Actinobacteriota, Fusobacteriota, Proteobacteria, Campylobacterota with no significant differences observed between healthy and allergic WHWT dogs. Previous studies including various dog breeds have demonstrated differences in fecal microbiota composition at the phylum level, with Firmicutes, Fusobacteriota, Bacteroidota, Proteobacteria, and Actinobacteriota being more prominent in healthy dogs, whereas Firmicutes, Bacteroidota, and Proteobacteria are predominate in allergic

Table 1. The dysbiosis index (DI) of healthy and allergic dogs. In addition to the DI, data on the individual bacterial taxa represented in the DI are shown for each dog.

| Dog No | Health status | Canine dysbiosis index* | Faecali-bacterium Log DNA | Turici-bacter Log DNA | Strepto-coccus Log DNA | E. coli Log DNA | Blautia Log DNA | Fusobac-terium Log DNA | Clostridium hiranonis Log DNA | Bifido-bacterium log DNA | Bacte-roides log DNA |
|---|---|---|---|---|---|---|---|---|---|---|---|
| 1 | allergic | −3.1 | 7.4 | 6.5 | 5.2 | 4.6 | 9.8 | 9.5 | 5.9 | 3.5 | 7.3 |
| 2 | allergic | −3.3 | 5.0 | 5.9 | 5.4 | 3.4 | 9.8 | 9.6 | 6.2 | 3.1 | 7.5 |
| 3 | allergic | 1.3 | 2.4 | 7.3 | 2.5 | 6.5 | 9.5 | 9.1 | 0.2 | 2.6 | 7.7 |
| 4 | allergic | −5.2 | 2.7 | 7.5 | 4.0 | 1.4 | 9.2 | 8.5 | 6.5 | 3.3 | 6.6 |
| 5 | allergic | −6.5 | 7.3 | 7.0 | 3.3 | 2.9 | 10.2 | 9.6 | 6.3 | 3.2 | 7.5 |
| 6 | allergic | −6.1 | 6.4 | 5.4 | 3.2 | 2.9 | 10.2 | 9.7 | 6.1 | 3.0 | 7.0 |
| 7 | allergic | −3.0 | 6.5 | 6.7 | 5.6 | 5.2 | 9.9 | 9.9 | 6.6 | 3.0 | 7.4 |
| 8 | allergic | −5.0 | 7.4 | 7.5 | 4.6 | 3.9 | 10.3 | 9.4 | 6.6 | 3.7 | 7.4 |
| 9 | allergic | −5.6 | 7.5 | 5.7 | 3.1 | 2.0 | 9.9 | 9.3 | 5.5 | 2.5 | 7.0 |
| 10 | allergic | −6.6 | 7.0 | 8.2 | 3.7 | 1.4 | 10.3 | 9.7 | 6.4 | 3.8 | 7.4 |
| 11 | allergic | −3.3 | 7.3 | 6.1 | 3.8 | 7.2 | 10.0 | 9.5 | 6.2 | 3.8 | 7.0 |
| 12 | allergic | −2.1 | 5.0 | 6.0 | 3.8 | 6.7 | 9.7 | 9.4 | 4.7 | 1.6 | 6.0 |
| 1 | healthy | −6.6 | 6.9 | 7.4 | 2.6 | 3.3 | 10.0 | 9.2 | 5.6 | 2.6 | 6.9 |
| 2 | healthy | −3.4 | 3.2 | 8.0 | 4.9 | 5.5 | 10.5 | 9.5 | 6.3 | 2.1 | 6.7 |
| 3 | healthy | −0.2 | 7.3 | 5.8 | 7.4 | 5.8 | 10.1 | 9.6 | 6.4 | 3.6 | 7.5 |
| 4 | healthy | −4.1 | 6.5 | 6.0 | 4.7 | 1.4 | 9.6 | 9.3 | 5.4 | 2.3 | 7.1 |
| 5 | healthy | −6.7 | 7.1 | 7.8 | 3.1 | 2.7 | 9.8 | 9.5 | 6.3 | 3.0 | 7.4 |
| 6 | healthy | 1.0 | 2.6 | 6.1 | 6.0 | 7.4 | 9.7 | 9.6 | 6.2 | 3.4 | 7.4 |
| 7 | healthy | −6.0 | 6.5 | 5.7 | 3.3 | 1.4 | 9.8 | 9.4 | 6.3 | 3.1 | 7.3 |
| 8 | healthy | −2.8 | 6.5 | 7.1 | 6.6 | 2.0 | 9.5 | 8.8 | 6.7 | 3.2 | 6.9 |
| 9 | healthy | −5.6 | 6.8 | 7.4 | 3.2 | 6.1 | 10.0 | 9.5 | 6.5 | 3.4 | 7.3 |

*A DI of < 0 is classified as normal dysbiosis index indicating that no shifts in the overall diversity of the intestinal microbiota have been detected. If individual bacterial groups are outside the reference interval, this is suggestive of minor changes. Values between 0 and 2 represent a mildly increased CI, suggesting a mild to moderate shift in the overall diversity of the intestinal microbiota. Values > 2 indicate that the DI is significantly increased, which is consistent with a major shift in the overall diversity of the intestinal microbiota [68]. The allergic dogs are presented in the table with blue shadow.

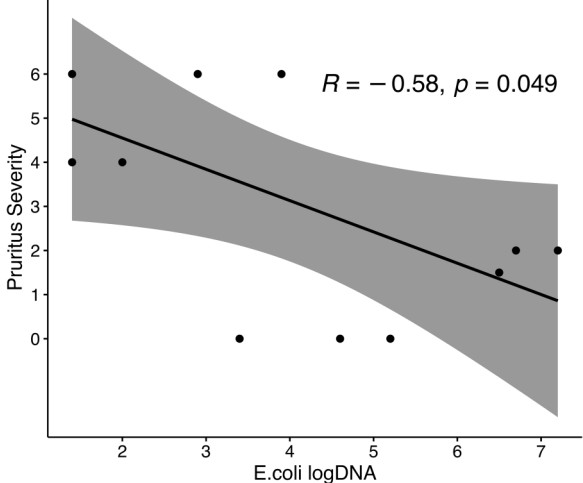

Fig 4. The correlation between absolute abundance of *E. coli* (log DNA) and the pruritus severity. *Escherichia coli* showed a moderate negative correlation with pruritus severity.

dogs [31]. The lack of a significant difference between the allergic and healthy population in this study may be explained by the breed-specific factors or because of the use of different methods in the sequencing analysis (DNA extraction, primers, bioinformatics pipelines) which all have. Furthermore, sequencing has high intra- and inter-laboratory variability, making comparison across studies using this approach alone difficult [35,36]. However, in this study we also examined important core bacterial taxa using qPCR, which is reproducible and allows comparison of overall microbiome shifts across previous studies, and we also did not observe any significant differences in microbial shifts and taxa [37].

Alterations in the gut microbiome have been linked to differences in allergen sensitivity and the development of atopic dermatitis in humans [21,38]. There is evidence, that the gut microbiota influences the tolerance towards allergens in many different ways, either directly interacting with T regulatory (Treg) cells or through the production of short-chain fatty acids (SCFA) such as butyrate, propionate, and acetate, which are exerting important advantageous immunomodulatory and anti-inflammatory effects both in the gastrointestinal tract and in the skin [39–41]. Furthermore, it was shown, that gut microbiota can exert an effect on post-translational host protein and epigenetic level [20]. Unfortunately, these aspects were not studied herein and should be addressed in future studies.

Abrahamsson et. al. showed that infants developing an atopic eczema had a lower intestinal microbial diversity [42]. Whereas in another study there were no significant differences found between infants with atopic dermatitis and healthy controls, but a difference could be identified at the compositional level. Infants with AD had a significantly lower abundance of Streptococcus [43]. Some human studies detected higher levels of Clostridium and Escherichia in the gut of infants with AD [44–46]. Whereas, in another study patients with AD had lower levels of Akkermansia, Bacteroidetes, and Bifidobacterium, compared to healthy controls [47,48]. Butyrate-producing bacteria (f.i. Coprococcus) were more abundant in healthy infants or infants with mild AD, compared to infants with severe AD in a study from Finland [49]. Based on this, one could hypothesize that maybe diversity alone may not be as important as specific microorganisms in the pathogenesis of AD [43]. However, it is important to note that findings from human medicine cannot be directly extrapolated to dogs. For instance, while Fusobacterium has been observed in healthy control dogs, it is associated with inflammatory bowel disease (IBD), a chronic gastrointestinal condition, of both, dogs and humans [50,51]. See Table 2 for a review of the current literature comparing to gut microbiota composition and diversity between allergic and healthy dogs.

It is noteworthy that many comparative studies have focused on the gut microbiome in children and infants. This early phase of life is crucial for the development of the immune system and represents a critical window that can promote the onset of allergic diseases in humans and in dogs [52–58]. During and after birth, we are exposed to a diverse world of bacteria coming from the mother and from the new environment which colonizes the gastrointestinal tract by oral ingestion [59–61]. Studies have shown that the colonization of the gut with bacteria increases significantly from birth and reaches a plateau around the age of three [60,62]. Thereafter, the gut microbiota stays relatively stable during adulthood and then in elderly people the diversity of the bacterial composition in the gut decreases [63]. The hygiene hypothesis posits that reduced exposure to pathogens and microbes during early life can lead to an increased risk of allergic diseases [53]. Several retrospective/cross sectional studies have demonstrated that growing up on a farm during childhood significantly reduces the risk of developing allergies and hay fever [64,65]. A prospective study from Finland showed that contact with farm animals played a crucial role in this protective effect [66].

To date all studies in allergic dogs, including the study herein, analyzed the gut microbiome in adult animals and not during the early life window of opportunity, which might explain the missing significant findings in this study. Future, longitudinal studies analysing the gut microbiome of puppies and following them into adulthood to determine early microbiome characteristics influencing the development of cAD are of paramount importance.

To assess if there is any relevant association between bacteria represented in the dysbiosis index and the pruritus severity, a correlation analysis was performed. A statistically significant negative correlation was observed between *E. coli* abundance and pruritus severity [49]. *E. coli*, a dominant species within the Enterobacteriaceae family, is a major component of the gut microbiota in humans and also in dogs [30,67,68]. Its rapid decline has been epidemiologically

**Table 2. A review of studies comparing the fecal metagenome between healthy dogs and dogs with canine atopic dermatitis.**

| Reference | Population | Sequencing data | Comparing diversities between allergic and healthy | | Comparing relative abundance between allergic and healthy dogs | |
|---|---|---|---|---|---|---|
| | | | Alpha diversity | Beta diversity | Significantly lower In allergic dogs | Significantly higher In allergic dogs |
| Thomsen, 2023 | 40 dogs | V3 - V4 region | Nonsignificant | Nonsignificant | Fusobacterium* | Escherichia/Shigella* |
| | 20 healthy | 16S rRNA | | | Megamonas* | Bacteroides |
| | 20 AD | | | | | Clostridium sensu stricto |
| | Shiba Inu | | | | | |
| Sugita, 2023 | 32 dogs | V3–V4 region | Significantly lower in AD | Significantly different | Fusobacterium* | Escherichia/Shigella* Ruminococcus gnavus group* |
| | 20 healthy | 16S rRNA | | | Megamonas* | Klebsiella |
| | 12 AD | | | | Prevotella | |
| | Various breeds | | | | Roseburia | |
| | | | | | Sutterella Phascolarctobacterium | |
| Rostaher, 2022 | 7 dogs | V3–V4 region 16S rRNA | Significantly lower in AD | Significantly different | Fusobacterium* | Conchiformibius Catenibacterium spp. |
| | 3 AD | | | | Lachnospira | Ruminococcus gnavus group* Megamonas |
| | 4 healthy | | | | Fecalibacterium | |
| | Beagle | | | | Ruminococcus torques group | |
| Uchiyama, 2022 | 25 dogs | V3–V4 region 16S rRNA gene | Nonsignificant, no clear tendency for differences | Significance almost reached | | Family Anaerovoracaceae |
| | 16 healthy | | | | | |
| | 9 AD | | | | | |
| | Shiba Inu | | | | | |

*Genera which are significantly different between healthy and allergic dogs in two or more studies.

associated with the onset of allergic diseases in humans [69]. Furthermore, in two studies with children of allergic mothers a preventive effect of early supplementation of newborns with *E. coli* strain-based probiotics on allergy development could be detected (EcO83) [70,71]. However, conflicting data exist as a higher *E. coli* abundance was found in individuals with atopic eczema compared to healthy individuals [44–46]. Furthermore, a positive correlation between the amount of *E. coli* and total serum IgE levels in infants with eczema is suggesting that higher levels of E. coli are associated with greater atopic sensitization [72]. The conflicting data can be attributed to several factors such as the heterogeneity of studies, missing population stratification, temporal microbiome dynamics and even measurement and analysis variability. Based on this newly discovered correlation between pruritus severity and *E. coli* abundance in WHWT dogs, further research is warranted to explore the molecular basis of this relationship in larger populations and in various dog breeds.

The lack of statistical significance observed in our study may be attributed several reasons. One could be a suboptimal phenotyping, which could have impacted the accuracy and reliability of group classifications. Healthy control dogs might in fact be affected dogs that did not yet express the disease phenotype. We attempted to minimize this by using healthy control dogs over the age of 5 years who had a greater chance of encountering the environmental trigger for the disease than a young dog, as previously suggested [11]. Another possibility could be, that our study was underpowered, as the disease prevalence was reported to be over 50% in the Swiss WHWT population [9]. A larger study with more dogs and with different breeds included would be helpful to clarify this question.

Lastly, it is also possible that the AD pathogenesis in WHWT is not mainly driven by shifts in the gut microbiota and that either specific microbial metabolites, altered interactions between mucosa-adherent microbiota and the immune system, and/or the genetics play a substantial role in not only shaping the gut microbiome in WHWT dogs but also influencing the development of AD through "non-microbiome" targets. Moreover, the pathogenesis of AD is very complex as it involves an interaction of many genes and environmental exposures affecting the immune and skin barrier thru different pathways and the effect of only one factor is likely to be relatively small [73]. For this, future studies should be including a systems biology approach, which will enable researchers to better understand this complex disease.

## Conclusions

The major findings of this study indicate that based on the assessment of fecal samples, shifts in the gut microbiota do not appear to be the primary driver in the pathogenesis of AD in WHWT dogs. However, we could find a moderate negative correlation with the amount of *E. coli*, which points towards a possible modulating effect on clinical symptoms. It is important to consider that this study was very likely underpowered, and future large-scale studies are necessary to validate these data and to explore the role of genetics and the gut microbiota across different dog breeds. Additionally, exploring the effects of early-life microbial exposure on microbiome composition and the development of allergic diseases could offer valuable insights into the prevention of AD, which is a highly relevant topic in current research.

## Supporting information

**S1 Fig. Relative abundance taxonomy bar plots at the class, order, family and genus level.**
(PDF)

## Author contributions

**Conceptualization:** Claude Favrot, Nina Maria Fischer, Ana Rostaher.

**Data curation:** Jan Suchodolski, Nina Maria Fischer, Ana Rostaher.

**Formal analysis:** Vera Felten, Jonathan Louis Turck, Jan Suchodolski, Nina Maria Fischer, Ana Rostaher.

**Funding acquisition:** Nina Maria Fischer, Ana Rostaher, Jan Suchodolski.

**Investigation:** Vera Felten, Jonathan Louis Turck, Claude Favrot, Nina Maria Fischer, Ana Rostaher.

**Methodology:** Vera Felten, Jonathan Louis Turck, Jan Suchodolski, Nina Maria Fischer, Ana Rostaher.

**Project administration:** Vera Felten, Jonathan Louis Turck, Claude Favrot, Jan Suchodolski, Nina Maria Fischer, Ana Rostaher.

**Resources:** Jonathan Louis Turck, Jan Suchodolski, Nina Maria Fischer, Ana Rostaher.

**Software:** Jan Suchodolski.

**Supervision:** Jonathan Louis Turck, Jan Suchodolski, Nina Maria Fischer, Ana Rostaher.

**Validation:** Vera Felten, Jonathan Louis Turck, Claude Favrot, Jan Suchodolski, Nina Maria Fischer, Ana Rostaher.

**Visualization:** Vera Felten, Jonathan Louis Turck.

**Writing – original draft:** Vera Felten, Nina Maria Fischer, Ana Rostaher, Jonathan Louis Turck.

**Writing – review & editing:** Vera Felten, Jonathan Louis Turck, Stefan Unterer, Claude Favrot, Jan Suchodolski, Nina Maria Fischer.

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
