## [Decision Letter · Decision Letter 0]

5 May 2025

Dear Dr. Rostaher,

We look forward to receiving your revised manuscript.

Kind regards,

Rajeev Singh

Academic Editor

PLOS ONE

**Journal Requirements:**

1. When submitting your revision, we need you to address these additional requirements. Please ensure that your manuscript meets PLOS ONE's style requirements, including those for file naming. The PLOS ONE style templates can be found at https://journals.plos.org/plosone/s/file?id=wjVg/PLOSOne_formatting_sample_main_body.pdf and https://journals.plos.org/plosone/s/file?id=ba62/PLOSOne_formatting_sample_title_authors_affiliations.pdf 2. We noticed you have some minor occurrence of overlapping text with the following previous publication(s), which needs to be addressed: https://www.sciencedirect.com/science/article/abs/pii/S0165242718302447?via%3Dihub In your revision ensure you cite all your sources (including your own works), and quote or rephrase any duplicated text outside the methods section. Further consideration is dependent on these concerns being addressed. 3. We note that the grant information you provided in the ‘Funding Information’ and ‘Financial Disclosure’ sections do not match.  When you resubmit, please ensure that you provide the correct grant numbers for the awards you received for your study in the ‘Funding Information’ section. 4. Thank you for stating the following financial disclosure: This study was funded by the Swiss National Science Foundation, American Kennel Club Canine Health Foundation and Westie Foundation of America  Please state what role the funders took in the study.  If the funders had no role, please state: "The funders had no role in study design, data collection and analysis, decision to publish, or preparation of the manuscript." If this statement is not correct you must amend it as needed. Please include this amended Role of Funder statement in your cover letter; we will change the online submission form on your behalf. 5. When completing the data availability statement of the submission form, you indicated that you will make your data available on acceptance. We strongly recommend all authors decide on a data sharing plan before acceptance, as the process can be lengthy and hold up publication timelines. Please note that, though access restrictions are acceptable now, your entire data will need to be made freely accessible if your manuscript is accepted for publication. This policy applies to all data except where public deposition would breach compliance with the protocol approved by your research ethics board. If you are unable to adhere to our open data policy, please kindly revise your statement to explain your reasoning and we will seek the editor's input on an exemption. Please be assured that, once you have provided your new statement, the assessment of your exemption will not hold up the peer review process. 6. Please upload a new copy of Figure 4 as the detail is not clear. Please follow the link for more information:  7.">https://blogs.plos.org/plos/2019/06/looking-good-tips-for-creating-your-plos-figures-graphics/" 7. Please include captions for your Supporting Information files at the end of your manuscript, and update any in-text citations to match accordingly. Please see our Supporting Information guidelines for more information: http://journals.plos.org/plosone/s/supporting-information.

Reviewers' comments:

Reviewer's Responses to Questions

**Comments to the Author**

1. Is the manuscript technically sound, and do the data support the conclusions?

Reviewer #1: Yes

Reviewer #2: Partly

2. Has the statistical analysis been performed appropriately and rigorously?

Reviewer #1: Yes

Reviewer #2: Yes

3. Have the authors made all data underlying the findings in their manuscript fully available?

Reviewer #1: Yes

Reviewer #2: Yes

4. Is the manuscript presented in an intelligible fashion and written in standard English?

Reviewer #1: Yes

Reviewer #2: Yes

**Reviewer #1:**  General comments:

Thank you for the opportunity to review this manuscript. Overall the manuscript was well written and provided novel information on a topic of high relevance to veterinarians in general and specialty practice, as well as those with interest in the link between the microbiome and states of health.

I have a few minor comments for the authors to consider:

Introduction:

- Introduction is a nice length, provides a concise yet comprehensive review of the relevant information to set up the study methods and findings.

Methods:

- Is the sample size a convenience sample size, or is there further justification for the chosen numbers of AD and control dogs? A comment on the sample size choice will be helpful.

- The dogs are largely a subset from the previously reported study of the birth WHWT cohort (previously published) that sampled dogs from several different litters/breeders. Were the WHWT dogs taken from the previously reported studies genetically related? Or were they mainly from separate litters and separate breeders?

- Were the samples batched and all analyzed at the same time point?

Results:

- To clarify (Line 196): treatments were administered prior to collection of the fecal sample? If so, were signs resolved (partially/completely) or still present at the time of sample collection?

- Ln 243-244: this sentence needs a minor grammatical correction (While no other…) as it seems incomplete as written.

- Ln 275: DI not CI?

Discussion:

- Ln 287-291: Are these studies all referencing gut microbiome, or any are referencing skin? I think it would be good to specify.

- Ln 310: there might be something missing from this sentence (ending in “all have.”)

- Table 2: I would hesitate to say ‘significance almost reached’ as there is no other context to interpret the findings outside of that comment (e.g. was the sample size too small, or other factors warranting discussion that could help support that claim? Without having space to discuss I don’t know if it is valid to use such a comment).

- Ln 346 paragraph: this is an interesting discussion point, but I would consider shortening / editing it emphasize your main points and more directly focus it on your study. From your statements I think the main things that are relevant to your study are (1.) early life microbiota changes would be missed by the current study design; (2) the discussion point on whether there are any studies to show that early life microbiota changes precede AD or other allergic diseases (?) For the references to allergies (# 64, 65) – are these references to allergies related to AD / skin / similar disorders? Or other types of allergies? I think the paragraph as it is currently written and referenced is a bit outside the scope of your discussion, so I’d like to see you bring it back to emphasize the main points of your findings.

**Reviewer #2: ** Authors present a well-structured and clearly written manuscript investigating the gut microbiota of healthy and allergic West Highland White Terrier dogs, with a high predisposition to canine atopic dermatitis (cAD). This study is timely and of scientific relevance, given increasing interest in the gut-skin axis and the potential role of gut dysbiosis in allergic diseases. The use of both DNA shotgun metagenomic sequencing and qPCR-based dysbiosis indexing is commendable and reflects a methodologically rigorous approach. The study is focused on a single breed, minimizing genetic and environmental variability, which is a notable strength. The integration of sequencing and qPCR analyses enhances the depth and reliability of the microbial profiling.

While the methodology is technically sound, the study includes only 21 animals (12 allergic, 9 healthy), which limits statistical power. The lack of significant differences in microbial diversity and composition between groups may reflect underpowering rather than a true absence of effect. This limitation is acknowledged by the authors but deserves further emphasis, especially in the conclusion.

The selection of healthy control dogs >5 years old is reasonable. However, given the high prevalence of cAD in WHWTs and the possibility of late-onset disease or subclinical presentations, the authors should discuss in more detail this bias.

The moderate negative correlation observed between E. coli abundance and pruritus severity is intriguing but difficult to interpret biologically. Given the conflicting evidence in human studies, the authors should temper their interpretation and clearly state that causality cannot be inferred. This could be expanded with a brief discussion of E. coli strain variability and its functional implications.

Although the authors used DNA shotgun sequencing, which allows for strain-level resolution, there is limited discussion on potential strain-specific roles or functional analysis. A short commentary on the limitations of taxonomy-based analysis versus functional inference (e.g., metagenomic pathways or SCFA synthesis potential) would enhance the depth of the discussion.

It would be helpful to clarify in the Methods section whether dogs receiving symptomatic allergy treatment were excluded from specific analyses, as treatment could impact microbial profiles.

**Do you want your identity to be public for this peer review?** For information about this choice, including consent withdrawal, please see our Privacy Policy

Reviewer #1: No

Reviewer #2: No

---

## [Author Response · Author response to Decision Letter 1]

24 Jun 2025

Reviewer #1: General comments:

Thank you for the opportunity to review this manuscript. Overall the manuscript was well written and provided novel information on a topic of high relevance to veterinarians in general and specialty practice, as well as those with interest in the link between the microbiome and states of health.

I have a few minor comments for the authors to consider:

Introduction:

- Introduction is a nice length, provides a concise yet comprehensive review of the relevant information to set up the study methods and findings.

Methods:

Question 1 : Is the sample size a convenience sample size, or is there further justification for the chosen numbers of AD and control dogs? A comment on the sample size choice will be helpful.

Answer: This is a very important question pointing out the main weakness of this study too low power. For this we now add a sentence in the materials and methods justifying the selected sample size and discuss this topic in depth in the discussion. Based on this fact our study can also deliver only preliminary data and this we write in the abstract so the reader is initially informed. Very likely we would need to have 50-100 dogs to detect an effect, which are now undertaking in a follow up study.

We write now in materials and methods (Line 98-101): “The sample size for this study was determined based on our previous research in Beagles, where we achieved significant results with fewer than 10 dogs [27]. By maintaining breed consistency, we aimed to minimize variability and enhance the robustness and reliability of our results.”

We make it also clear in the abstract that this study is preliminary (line 6). “This preliminary study aimed to compare the gut microbiome of allergic and healthy WHWT dogs to explore its role in cAD.»

We added also a clearer comment on this limitation in the discussion (Line 421-427): “Another possibility could be, that our study was underpowered. The relatively small sample size allowed us to detect only large effect sizes. Given that AD is a complex disease with multiple contributing factors, including the microbiome, one should anticipate that microbial effects on AD pathogenesis are likely to be of smaller magnitude. Future studies should aim for larger sample sizes, especially in high-risk breeds such as WHWT with AD prevalence up to 50% [9], to detect these small effect sizes.”

Question 2: The dogs are largely a subset from the previously reported study of the birth WHWT cohort (previously published) that sampled dogs from several different litters/breeders. Were the WHWT dogs taken from the previously reported studies genetically related? Or were they mainly from separate litters and separate breeders?

Answer: The dogs were mainly from different breeders and were not genetically related. We now write the following sentence(Line 88-89): “Most of the dogs in this study were not genetically related and were not from the same breeder.”

Question 3: Were the samples batched and all analyzed at the same time point?

Answer: Thank you for this comment. Yes, all were batch analyzed. We thought it was clear in the initial manuscript as we mentioned that we first collected all samples and then shipped them to the lab for analysis. To clarify this, we added a sentence: »The fecal samples were collected from both groups and were stored at -80 °C before being shipped on dry ice to the Gastrointestinal Laboratory of Texas A&M University, College Station, Texas (batch analysis of all samples).»

Results:

Question 4: To clarify (Line 196): treatments were administered prior to collection of the fecal sample? If so, were signs resolved (partially/completely) or still present at the time of sample collection?

Answer: Only 3 of the 12 AD dogs where currently under treatment but still exhibited clinical signs. To make this clear we added into the results section the following sentence (Line 203-204): “These three dogs still exhibited clinical signs of AD despite symptomatic treatment.”

Question 5: Ln 243-244: this sentence needs a minor grammatical correction (While no other…) as it seems incomplete as written.

Answer: Thank you for this comment. We improved the sentence as follows (Line 255-256): “The remaining dogs in both groups did not display a bacterial shift indicative of gut dysbiosis.»

Question 6: Ln 275: DI not CI?

Answer: Thank you, we changed it.

Discussion:

Question 7: Ln 287-291: Are these studies all referencing gut microbiome, or any are referencing skin? I think it would be good to specify.

Answer: Thank you for this comment. We specified that these studies analyzed the gut microbiome. We changed the sentence as follows (Line 315-317): “Previous studies have shown significant differences in alpha and beta diversity of the gut microbiome between healthy and allergic dogs of various breeds [31] and in Beagle dogs [27], with allergic dogs typically exhibiting lower alpha diversity.”

Question 8: Ln 310: there might be something missing from this sentence (ending in “all have.”)

Answer: Thank you for pointing this out. By mistake we deleted a part of the sentence. Now we write (Line 336-339): “The lack of a significant difference between the allergic and healthy population in this study may be explained by the breed-specific factors or because of the use of different methods in the sequencing analysis (DNA extraction, primers, bioinformatics pipelines).”

Question 9: Table 2: I would hesitate to say ‘significance almost reached’ as there is no other context to interpret the findings outside of that comment (e.g. was the sample size too small, or other factors warranting discussion that could help support that claim? Without having space to discuss I don’t know if it is valid to use such a comment).

Answer: Thank you so much for this valuable comment. We removed this sentence and say it was nonsignificant (see Table 2).

Question 10: Ln 346 paragraph: this is an interesting discussion point, but I would consider shortening / editing it emphasize your main points and more directly focus it on your study. From your statements I think the main things that are relevant to your study are (1.) early life microbiota changes would be missed by the current study design; (2) the discussion point on whether there are any studies to show that early life microbiota changes precede AD or other allergic diseases (?) For the references to allergies (# 64, 65) – are these references to allergies related to AD / skin / similar disorders? Or other types of allergies? I think the paragraph as it is currently written and referenced is a bit outside the scope of your discussion, so I’d like to see you bring it back to emphasize the main points of your findings.

Answer: Thank you so much, this is true, the “flow” of this paragraph was not optimal. We tried to improve it. The main thing we want to say is, as you already pointed out, that the early life gut microbiota is essential for disease development and that unfortunately we did not address this in the current study. Regarding your question about Reference 64 and 65, these studies focused on asthma and hay fever. However, these are seminal studies that contributed to the development of the hygiene hypothesis, which was the basis that we arrived at the point where we are now: studying the gut microbiome in relating to allergies and other chronic/immunologic diseases. For this reason, we would leave these references but appropriately mention the allergy type they refer to. However, considering you valuable advice, we shortened and adapted the paragraph according to this discussion.

Here you can read the revised version of this paragraph (Line 379-391): “Research related to the hygiene hypothesis has emphasized the microbiome's role in several immune-mediated including allergies, showing its influence not only on the disease progression but also its development [52-58]. For example, farm upbringing in childhood reduces asthma and hay fever risk, with farm animal contact being protective [59-61]. The first 1000 days of life are fundamental in the establishment of a healthy immune system and also gut microbiome, whereby several factors can influence its normal composition and function [62] . In this period, unhealthy cues can cause alterations in gene expression later in life, increasing the risk of multifactorial environmentally driven diseases such as allergies [63]. Current studies on allergic dogs, including this one, focus on adult microbiomes, missing the early life gut microbiome. Future, longitudinal studies analyzing the gut microbiome of puppies and following them into adulthood to determine early microbiome characteristics influencing the development of cAD are of paramount importance.”

Reviewer #2: Authors present a well-structured and clearly written manuscript investigating the gut microbiota of healthy and allergic West Highland White Terrier dogs, with a high predisposition to canine atopic dermatitis (cAD). This study is timely and of scientific relevance, given increasing interest in the gut-skin axis and the potential role of gut dysbiosis in allergic diseases. The use of both DNA shotgun metagenomic sequencing and qPCR-based dysbiosis indexing is commendable and reflects a methodologically rigorous approach. The study is focused on a single breed, minimizing genetic and environmental variability, which is a notable strength. The integration of sequencing and qPCR analyses enhances the depth and reliability of the microbial profiling.

Comment 1: While the methodology is technically sound, the study includes only 21 animals (12 allergic, 9 healthy), which limits statistical power. The lack of significant differences in microbial diversity and composition between groups may reflect underpowering rather than a true absence of effect. This limitation is acknowledged by the authors but deserves further emphasis, especially in the conclusion.

Answer:

Thank you for this very valuable comment. We added a paragraph in the discussion (Line 421-427): “Another possibility could be, that our study was underpowered. The relatively small sample size allowed us to detect only large effect sizes. Given that AD is a complex disease with multiple contributing factors, including the microbiome, one should anticipate that microbial effects on AD pathogenesis are likely to be of smaller magnitude. Future studies should aim for larger sample sizes, especially in high-risk breeds such as WHWT with AD prevalence up to 50% [9], to detect these small effect sizes.“

We also added a reasoning for this sample size. We write now in materials and methods (Line 98-101): “The sample size for this study was determined based on our previous research in Beagles27, where we achieved significant results with fewer than 10 dogs. By maintaining breed consistency, we aimed to minimize variability and enhance the robustness and reliability of our results.”

Comment 2: The selection of healthy control dogs >5 years old is reasonable. However, given the high prevalence of cAD in WHWTs and the possibility of late-onset disease or subclinical presentations, the authors should discuss in more detail this bias.

Answer:

Atopic dermatitis usually starts before the age of 3 years so 5 years is a much more stringent criterium. Furthermore, a birth cohort study with 108 WTWT dogs showed that the disease prevalence was 52% by the age of 3 years, which is not only reaching but also exceeding the estimated prevalence for AD. Based on these data we think that the 5 years are good choice to secure the inclusion of only healthy animals. To clarify this, we now added a sentence in the discussion (Line 419-421): “Furthermore, a birth cohort study in WHWT dogs showed that the expected AD prevalence in this breed was already reached or even exceeded by the age of 3 years [9].”

Comment 3: The moderate negative correlation observed between E. coli abundance and pruritus severity is intriguing but difficult to interpret biologically. Given the conflicting evidence in human studies, the authors should temper their interpretation and clearly state that causality cannot be inferred. This could be expanded with a brief discussion of E. coli strain variability and its functional implications.

Answer: We appreciate this important suggestion. We have revised the discussion section to clearly state that causality cannot be inferred from our results. Furthermore, we now include a short paragraph discussing the variability among E. coli strains and their differing functional roles. The revised text reads as follows (Line 406-411): “It is important to note that Escherichia coli represents a highly diverse group of strains with considerable genetic and functional variability (65) While some strains are commensal and part of a healthy gut microbiome, others may possess pathogenic traits (70). As our analysis did not resolve strain-level taxonomy, no conclusions about the potential role or function of E. coli in the disease context of cAD can be drawn at this stage.”

Comment 4: Although the authors used DNA shotgun sequencing, which allows for strain-level resolution, there is limited discussion on potential strain-specific roles or functional analysis. A short commentary on the limitations of taxonomy-based analysis versus functional inference (e.g., metagenomic pathways or SCFA synthesis potential) would enhance the depth of the discussion.

Answer: We appreciate your suggestion to enhance the discussion on the limitations of taxonomy-based analysis versus functional inference. Therefore, we added the following sentence to the discussion (Line 436-442): In our study, while DNA shotgun sequencing provided strain-level resolution, we recognize the limitations of taxonomy-based analysis in fully capturing the functional potential of the gut microbiome. Future research should integrate functional analyses, such as metagenomic pathways and short-chain fatty acid synthesis potential, to better understand the specific roles of microbial taxa in the pathogenesis of canine atopic dermatitis. This approach will enhance the insights into how the gut microbiome influences health and disease in dogs.

Comment 5: It would be helpful to clarify in the Methods section whether dogs receiving symptomatic allergy treatment were excluded from specific analyses, as treatment could impact microbial profiles.

Answer: Thank you so much for this question. Only one dog from these 12 allergic received oral drugs (Apoquel), therefore we do not believe that this could have a major impact on the results. The reasoning why it was legitim to include this dog:

1. He still showed clinical signs of AD

2. A recent study showed that Apoquel seems not to affect the gut microbiome in allergic dogs: https://www.mdpi.com/2076-2615/12/18/2377

We agree with you that this information is important for the article, we added two sentences to the materials & methods section (line 94-97): “Dogs were allowed to receive orally only the symptomatic treatment oclacitinib, which was previously shown not to affect the gut microbiota [27]. Dogs receiving antibiotics within the last 6 months were excluded.

---

## [Editor Report · Decision Letter 1]

26 Jun 2025

An insight into the gut microbiota of healthy and allergic West Highland Whiter Terrier dogs

PONE-D-25-07058R1

Dear Dr. Rostaher,

We’re pleased to inform you that your manuscript has been judged scientifically suitable for publication and will be formally accepted for publication once it meets all outstanding technical requirements.

Kind regards,

Rajeev Singh

Academic Editor

PLOS ONE
---

## [Editor Report · Acceptance letter]

PONE-D-25-07058R1

PLOS ONE

Dear Dr. Rostaher,

I'm pleased to inform you that your manuscript has been deemed suitable for publication in PLOS ONE. Congratulations! Your manuscript is now being handed over to our production team.

Kind regards,

on behalf of

Dr. Rajeev Singh

Academic Editor

PLOS ONE